# The Middle East and Africa Code of Promotional Practices in the Pharmaceutical Industry

**Sand Salhout [1] and Clemens Bechter [2,*]**

[1] Dual Studies Business Administration Program, Faculty of Business and Economics, Al-Quds University, Abu Dis 20002, Palestine; ssalhout@staff.alquds.edu

[2] Thammasat Business School, Tha Prachan, Bangkok 10200, Thailand

[*] Correspondence: clemensbechter@tbs.tu.ac.th

**Abstract:** The pharmaceutical industry is known for investing heavily in promotions targeted at healthcare professionals (HCPs). Governments around the world try to regulate unwanted promotional practices in different ways. Where binding laws are in place in the U.S.A., European governments favor self-regulation. The purpose of this research is the evaluation of the Middle East and Africa Code of Promotional Practices (MEACPP) as a preliminary draft and its implications. Our paper fills a research gap by looking into the perceptions of the parties involved, analyzing their interests, and predicting possible outcomes. We used a mixed-method approach. Interviews were conducted with pharmaceutical companies and associations; while a questionnaire was administered to HCPs. Our findings suggest that all parties are in favor of more transparency. However, when it comes to disclosing the received financial support, the HCPs are hesitant. An estimated 20% would be willing to fully disclose their received benefits, which is in line with their European colleagues. Multinational pharmaceutical companies follow their own in-house standards and fear being at a competitive disadvantage when local companies can promote their drugs without any strings attached. MEA pharmaceutical companies do not see the potential benefits of analyzing the publicly available data to identify key opinion leaders (KOLs). The limitation of our research is the fact that the MEACPP has not been implemented yet and survey results are therefore based on expectations rather than real events.

**Keywords:** Middle East and Africa Code of Promotional Practices; pharmaceutical industry; transparency code; self-regulation

## 1. Introduction

Most people would agree that corruption is a bad thing. However, the pharmaceutical industry is doing just that by incentivizing healthcare professionals (HCP) such as doctors, and healthcare organizations (HCOs) such as hospitals, to prescribe their drugs. Incentives come in various forms such as gifts, invitations to seminars, sponsored golf tournaments (in combination with workshops), and weekend 'fact-finding missions' to scenic cities, etc. A study from ProPublica found that HCPs who accepted payments in 2014 were two to three times more likely to prescribe high rates of brand-name drugs compared with others (Jones and Ornstein 2016). Findings demonstrated that the amount of payments correlated with the volume of brand-name prescriptions. Dejong et al. (2016) found that receipt of sponsored hospitality was associated with an increased rate of prescribing brand-named medicine. The global pharmaceutical's market is forecasted to have a value of US $1294 billion by 2021, which is an increase of 36.8% since 2016 (Global Pharmaceuticals Industry 2017). The U.S.A. accounts for around 40% of the global pharmaceutical's market value, while the Middle East and North Africa (MENA) are around 5%.

Most governments and pharmaceutical associations are aware of unethical practices and try to regulate promote activities in the pharmaceutical industry.

Basically, there are three ways a country can restrict promotional activities:

1. Law
2. Self-Regulation
3. Watchdog

Restrictions imposed by law are expensive to monitor and limit the freedom of conducting business. Self-regulation, on the other hand, is often used to pre-empt a law and is a cheaper way. Many industries make use of self-regulation; the food industry is one example where this seems to be working (Jensen and Ronit 2015). The tobacco and alcohol industries, on the other hand, are regulated by law in almost all countries, for example, fighting alcohol consumption (Savell et al. 2016) and electric cigarettes targeting teenagers (Chaffee et al. 2018). However, it is not always effective. Government Watchdogs have no real power, as the name suggests, they just watch. This is the weakest form of control.

Regulations for promotions in the pharmaceutical industry differ around the globe. In the U.S.A., promotions are regulated by law, and in Europe, Australia, Brazil, and Japan, by self-regulation. In the Middle East and Africa, there is no binding regulation yet. A first attempt is The Middle East and Africa Code of Promotional Practices in the Pharmaceutical Industry (MEACPP).

In 2010, the U.S.A. enacted a law called 'The Physician Payment Sunshine Act' (PPSA), which forces pharmaceutical companies to disclose all promotional activities and prohibits expensive gifts altogether (Hwong et al. 2014).

The PPSA attracted lots of attention from news media as well as academic journals. Toroser et al. (2016) found that among 1200 indexed publications between January 2010 and June 2015, 113 had content on the PPSA. Whether the law achieved its goals depends on the point of view.

A study by Karas et al. (2017) found that 42% of US pediatricians received payments from pharmaceutical companies; 35,697 pediatricians received US $30,031,960, which translates into an average of US $840 per head in 2014. Orthopedic surgeons were amongst the top sponsored with a mean payment of US $7261, and US$ 7,849,711 is the largest payment to an individual orthopedic surgeon (Lopez et al. 2016). In comparison, the highest amount that a transplant surgeon received was only US $83,520 (Ahmed et al. 2016). A survey among 1987 US patients showed that only 12% of survey respondents knew that HCP's payment information was publicly available and only 5% knew whether their own doctor had received payments (Pham-Kanter et al. 2017). To increase this low percentage, Young et al. (2018) suggested that HCPs should (preferably must) post a hyperlink on their website to the public database where payments are listed, i.e., a patient can find out easily how much money their own doctor received.

One can conclude that patients are not well informed and the PPSA has not stopped pharmaceutical companies from spending money on HCPs. The only advantage might be that politicians can more closely examine the linkage between the pharmaceutical industry spending directed at HCPs and their prescription pattern and then intervene. However, no such action has ever been taken. Relating prescribed drugs to payments made by pharmaceutical companies would be too simplistic. There are many factors that influence prescriptions. For example, Mano-Negrin and Mittman (2001) point out that clinical behavior also depends on peer group social networks. It may also depend on the HCP-patient relationship. Applying an equity theory may show which pharmaceuticals emotionally exhausted HCPs preference (vice versa) and the reasons for it (Van Dierendonck et al. 1994).

In Europe, countries opted for industry measures that require the voluntary disclosure of payments and gifts to physicians. The 'European Federation of Pharmaceutical Industries and Associations' (EFPIA) disclosure code tries to guarantee transparency in the pharmaceutical industry in a way that payments to HCPs/HCOs must be reported and are publicly available the following year (Buske et al. 2016). Each European country is free to implement its own version of the disclosure code; however, they are all in line with the EFPIA code, so it is more of a question of governance than content.

For example, in Germany, it is called 'Transparenzkodex' (transparency code), which is overseen by the association FSA (Freiwillige Selbstkontrolle für die Arzneimittelindustrie e.V.). In France, the code is overseen by the French Drug Agency ASNM, in Italy by the Italian Medicines Agency AIFA, in Spain by the Asociación Nacional Empresarial de la Industria Farmacéutica (Farmaindustria), and in the UK by ABPI (Association of the British Pharmaceutical Industry).

Whether by law or by self-regulation, there are four main categories of financial transactions involving HCPs and HCOs that must be reported:

(1) R&D projects
(2) Donations
(3) Support for seminars and workshops (travel, accommodation, and registration fees etc.)
(4) Consulting fees.

The overall goal is to make payments to HCPs/HCOs more transparent; meanwhile, data protection laws must be observed. However, this is a very controversial legal issue. Data protection laws, around the world, protect citizens and not companies. This makes an important difference. Some argue that an HCO is a private person; others say that she/he is a practitioner, and as such, a company, and therefore, cannot rely on data protection reasons to escape disclosure. She/he receives the money for professional work and not as a private individual. As mentioned, the issue is legally highly controversial (Michael 2016).

In Germany, any interested person can see which pharmaceutical company gave how much money, for which purposes, to an HCP/HCO. There are even market research companies that analyze these data and recommend which doctors to target. Looking at the payments received, it is easy to identify key opinion leaders (KOLs) in a therapeutic area. Publicly available information includes the name and address of the HCP/HCO and the amount of money received, split by category. There are publicly available sources where patients can see the amount of money received by their doctors, see Figure 1.

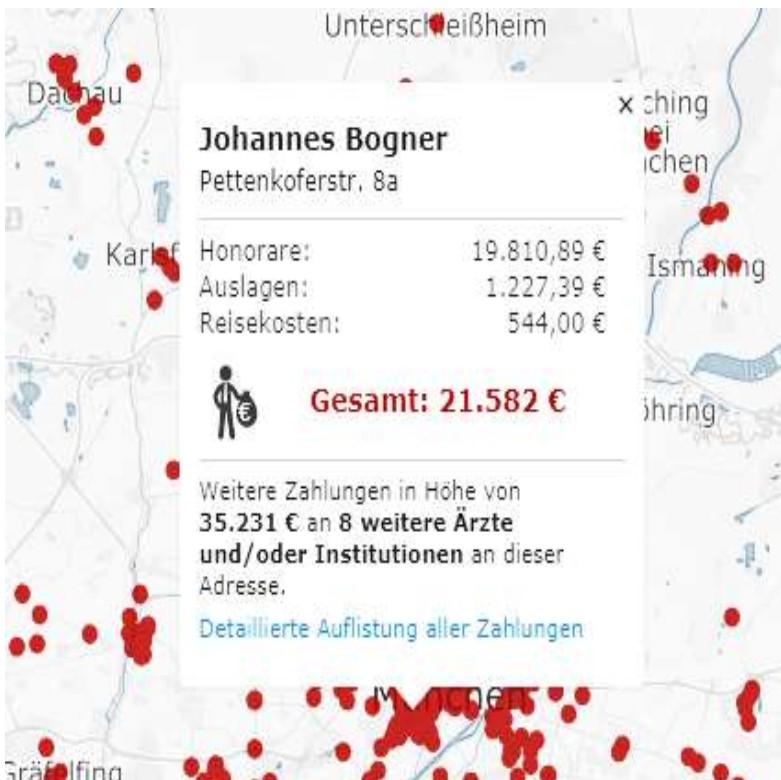

**Figure 1.** Sample report, payments received by HCP (here: Dr. Johannes Bogner, Munich).

Each dot represents an HCP. On the interactive map, anyone can search for an HCP and the payments she/he received. In the above case, the HCP received a total of EUR 21,582 in 2016, out of which 19,810 were consulting fees, and 1227 & 544 in support for seminars and workshops. A further EUR 35,231 went to another 8 HCPs at the same address.

Participating in the German transparency code is voluntary, although strongly enforced by the pharmaceutical association FSA. In 2015, only 31% of HCOs and HCPs participated in disclosing their income from pharmaceutical companies. In 2016, the value decreased to 25%, and in 2017 there was a further decrease to 20%. The reported figure, and most likely not the true figure, of payments to HCPs and HCOs for 2017 in Germany was EUR 605 million, compared to EUR 562 million in 2016. In 2017, only 55 companies representing 75% of the German market reported their payments to HCPs and HCOs. Therefore, one can guess that the true amount might be closer to EUR 1 billion p.a. in Germany alone. The estimated figure for the USA is US $8 billion for 2017.

Research has shown that between 2004 and 2012 there was an average of more than one code breach a week in Sweden and the UK (Lo 2015). EFPIA argued that this is a "testimony to the efficiency of these self-regulatory systems that misconduct is indeed identified" (Lo 2015). There are always two interpretations to a story.

The Middle East and Africa Code of Promotional Practices (MEACPP, the 'code'), finalized in 2011, is a set of guiding principles that were developed and should be enforced by pharmaceutical companies to self-regulate marketing and promotional practices. The code covers 32 pages[1]).

The code is in line with the provisions of the International Federation of Pharmaceutical Manufacturers Associations (IFPMA) Code of Pharmaceutical Marketing Practices and the European Federation of Pharmaceutical Industries and Associations (EFPIA).

Important regulations are:

Section 4.01. "Data privacy of healthcare professionals should be observed."

Section 5.01. "Programs shall not disguise or misrepresent their true promotional intent. Examples include market research studies and programs intended to promote a specific product."

Section 11.01. "No gift, pecuniary advantage or benefit in kind may be supplied, offered or promised to a healthcare professional as an inducement to prescribe, supply, sell or administer a medicinal product."

Section 14.09. "Each company must appoint at least one senior employee who shall be responsible for supervising the company and its subsidiaries to ensure that the standards of the applicable Code(s) are met."

The code looks comprehensive, but falls short of demanding pharmaceutical companies to publish the amount of money that went to HCPs and HCOs. This is the main difference between the MEACPP code and US law or the European self-regulation that both demand publishing these figures. In general, regulation in MEA pharmaceutical industry is still scarce (Ball and Fallows 2017).

Only a few MEA countries have set up task forces and initiated first steps:

Egypt: PhRMA's Egypt Chair oversees the promotional activities in the pharmaceutical sector.

South Africa: The Ministerial Procurement Task Team (MPTT) oversees the procurement of pharmaceuticals, however, only for the public sector. In addition, a corruption hotline has been installed.

Saudi-Arabia has launched the New Saudi Promotional Code by the Saudi FDA (SFDA). The Code covers 11 articles with approximately one page per article. The Saudi Chamber of Commerce in cooperation with SFDA will monitor the implementation.

Tunisia: The SEPHIRE (le Syndicate des Entreprises Pharmaceutiques Innovantes de Recherche) oversees promotional standards for biotech companies and research-based pharmaceutical companies.

UAE is probably the most advanced country in respect to the implementation of promotional standards (Emerald Group Publishing Limited 2013).

---

[1]  http://www.dgxdemo.com/phrmag/wp-content/uploads/2017/06/MEA-ENG-CODE-FINAL.pdf.

The above list shows that the implementation of the code is still lacking. Only a few countries have done the first step by appointing a responsible institution for the forthcoming implementation.

Another indicator for the need to disseminate the code is the Google search result for the term "Middle East and Africa Code of Promotional Practices", which returns only 11 relevant result pages that are non-academic in nature. EBSCO, Google Scholar, SCOPUS, and Web of Science return no articles at all on the query!

The objectives of the study are to:

Assess the willingness of HCPs to allow pharmaceutical companies to publish personalized data on the financial support given.

Estimate how many years it will take to get the MEACPP implemented.

Evaluate the pros and cons of a law versus self-regulation.

Predict which model MEA will adapt and the likely outcomes.

Figure 2 shows the research framework. HCPs might challenge binding regulations in court by citing data protection reasons. The pharmaceutical industry will try to mold any code in their favor. In contrast, governments may want to see stricter rules applied. The third research question addresses the point of an information audit. By evaluating publicly available data, a company can assess competitive actions (Barker 1990). Identifying target groups and individual KOLs is now possible.

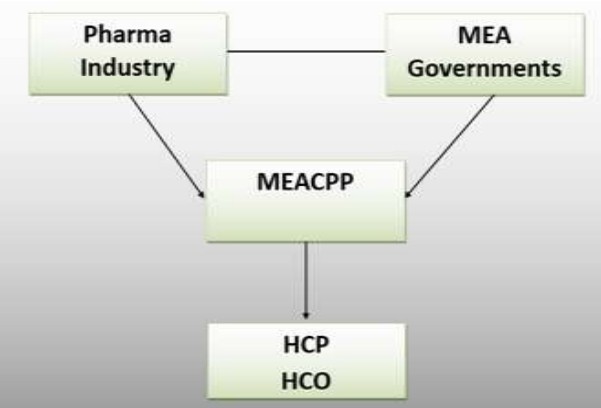

**Figure 2.** Research Framework.

## 2. Materials and Methods

There are several players in the pharmaceutical industry: HCPs, HCOs, pharmaceutical companies, associations, and health ministry, etc. The pharmaceutical industry is more dominated by strategic alliances than rivalry (Yoon et al. 2018). One reason might be that margins for branded products are high, and the fact that the buyer (HCP) and the consumer (patient) are in most cases not paying at all, or just a fraction for their medication. Costs of pharmaceutical products continue to rise, as does pressure to adopt direct or indirect controls on pharmaceutical prices.

Fabbri et al. (2018) conducted a descriptive content analysis of the transparency provisions implemented by February 2017 in nine European Union (EU) countries concerning payments to HCPs, they found that reporting is often incomplete or is difficult to analyze the file formats, such as pdf, and common influential gifts such as food and drinks are excluded.

The perceptions of HCPs and HCOs will play a major part in the acceptance of MEACPP and future additions that might include financial disclosure.

Our research questions are:

- Do pharmaceutical companies see a threat in MEACPP in its current form?
- If financial disclosure was part of the code, will HCPs participate or cite data protection reasons for not disclosing their data?
- To what extent would pharmaceutical companies use the publicly available data to identify KOLs?

Our research hypotheses are that: (1) HCPs will oppose the idea of publicizing their received contributions; (2) Pharmaceutical companies will also not be enthused by the idea of making their payments transparent; (3) Pharmaceutical associations will show that they support transparency in order to avoid legal sanctions, but being representative of the industry, they will try to keep public disclosure of financial data to a minimum; (4) Patients will be interested in retrieving the amount of contributions their practitioner received; and (5) Health Ministries will be the driving force in pushing transparency codes forward.

The authors used a mixed-method approach (Creswell 2009 and Dowding 2013). We interviewed health ministries as well as people who were part of the initial drafting of the MEACPP. In addition, we corresponded with authors who have published on the Sunshine Act (U.S.A.) and the German transparency code. Based on these interviews, we developed our quantitative questionnaire. Since there was no existing questionnaire, we developed our own based on these interviews.

HCPs filled in a questionnaire. The four pharmaceutical companies interviewed represent companies that are present in more or less all MEA countries. The 61 HCPs are all from the Middle East and are all Arabs. In addition, one hospital manager was interviewed. The review of the status quo has shown that the most advanced countries for our research are all from the ME, with only one exception: South Africa. We, therefore, focused our research on the Middle East. The survey was posted online between the middle of June and the end of July 2018. We also interviewed five senior faculty at ME universities; however, they have never heard about MEACPP, so the interviews were ineffective. The interview guidelines and the questionnaire are posted in the Appendix. The questions were phrased in a predictive analytics manner, e.g., 'would you as an HCP be willing to disclose the payments that you have received from pharmaceutical companies'? Out of the 61 respondents, 42 were male and nine female. Their medical disciplines are represented in Table 1. We used convenience sampling whereby HCPs were encouraged to spread the word about the survey among their friends.

**Table 1.** Sample demographics—jobs.

|  | Frequency | Percent | Valid Percent | Cumulative |
|---|---|---|---|---|
| Gynecology | 16 | 26.2 | 26.2 | 26.2 |
| Emergency | 11 | 18.0 | 18.0 | 44.3 |
| Internal Medicine | 11 | 18.0 | 18.0 | 62.3 |
| General Medicine | 10 | 16.4 | 16.4 | 78.7 |
| Cardiology | 6 | 9.8 | 9.8 | 88.5 |
| Pediatrics | 4 | 6.6 | 6.6 | 95.1 |
| Dentist | 2 | 3.3 | 3.3 | 98.4 |
| Surgery | 1 | 1.6 | 1.6 | 100.0 |
| Total | 61 | 100.0 | 100.0 |  |

In most countries, the top three medical disciplines are General Medicine, Internal Medicine, and Pediatrics. However, we feel that the discipline does not have an impact on the survey results. Any profession uses medication sold by pharmaceutical companies. The real question is less about the discipline, but rather the volume that a doctor is prescribing.

## 3. Results

### 3.1. HCPs and HCOs

As seen in Figure 3, more than 40% of HCPs believe that the MEACPP will never be implemented. Another 20% are more positive and think that it will happen within the next 5 years.

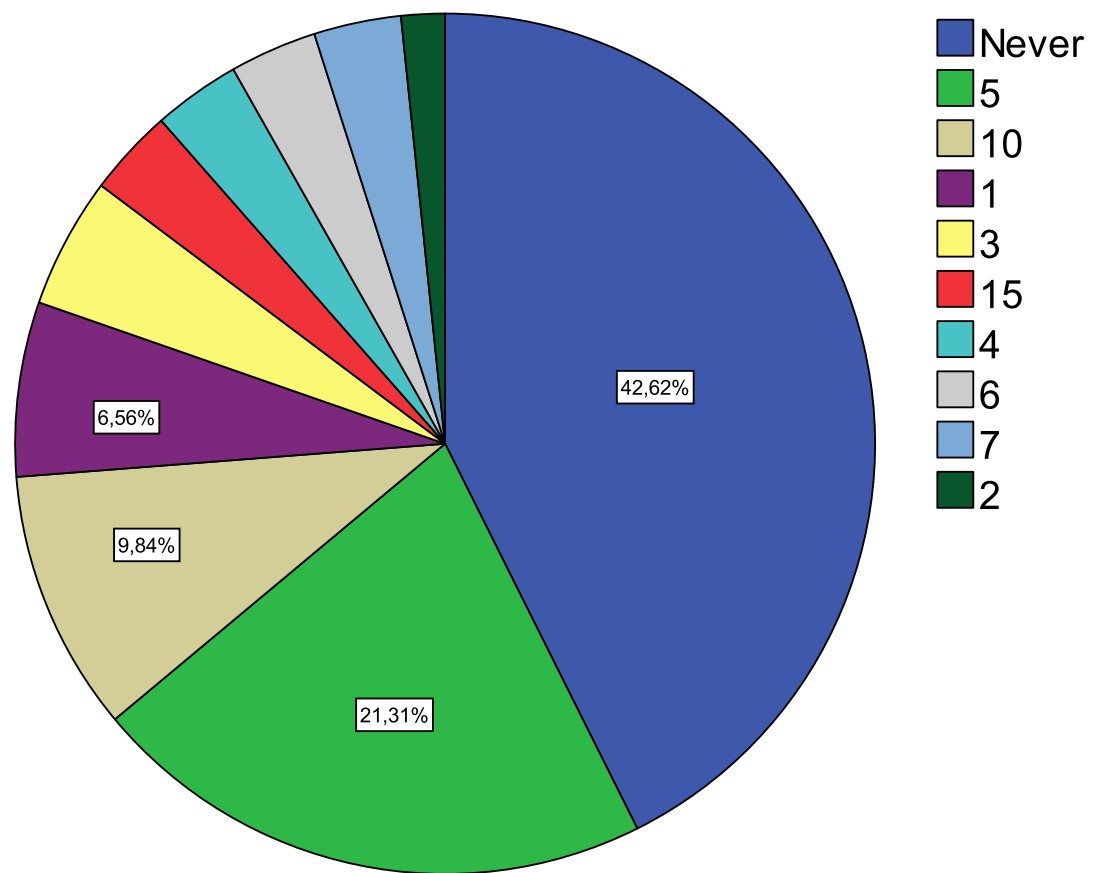

**Figure 3.** Years to implement.

The result is somewhat surprising when considering that the U.S.A. and Europe have similar codes in place already; rather, it appears to be a question of when and not how.

Another surprising result is the perception that the pharmaceutical industry as a whole is perceived as corrupt, with an average of 4.08 on a one (totally disagree) to five (totally agree) Likert scale (see Table 2). The HCPs are part of the industry; thus, if anyone is the beneficiary of such corruption, then it is the HCPs themselves.

**Table 2.** Descriptive Statistics.

|  | *N* | **Mean** | **Std. Deviation** |
|---|---|---|---|
| Transparency | 61 | 4.23 | 0.783 |
| Corrupt | 61 | 4.08 | 0.881 |
| Peers | 61 | 3.77 | 0.990 |
| Data Protection | 61 | 3.33 | 1.136 |
| Patient Interest | 61 | 2.95 | 1.102 |
| Valid *N* | 61 |  |  |

HCPs also agree that such a code would be a good thing to have because it guarantees transparency. However, this relates to the publication of payments made by pharmaceutical companies and not the declaration of the beneficiary. The high standard deviation of the answer, whether HCPs would cite data protection reasons for not having to disclose their received contributions, shows mixed opinions on the matter. Seeing what their colleagues (peers) are earning is of moderate interest to HCPs.

HCPs think that the fact that patients can now see how much an HCP received will not increase the patient's trust. This can be interpreted in two ways: Either the trust between HCPs and the patient is already very high and cannot be improved further, or the code itself will not make any difference.

One can conclude that HCPs like the idea of such a transparency code. However, they prefer a system whereby the pharmaceutical industry publishes payments without naming the HCP who received the money.

The question is whether they would oppose the publication of their data by referring to data protection. As shown in Table 3, only 26.2% strongly disagree or disagree. In comparison, the German percentage was 31% when the transparency code started in 2015 and now stands at 20%. One can assume that the percentage of 26.2% in MEA will decrease rather than increase over time. It may well stabilize at around 20% long-term.

**Table 3.** Data Protection.

| Answer | Frequency | Percent | Valid Percent | Cumulative |
|---|---|---|---|---|
| Totally Disagree | 3 | 4.9 | 4.9 | 4.9 |
| Disagree | 13 | 21.3 | 21.3 | 26.2 |
| Neutral | 16 | 26.2 | 26.2 | 52.5 |
| Agree | 19 | 31.1 | 31.1 | 83.6 |
| Totally Agree | 10 | 16.4 | 16.4 | 100.0 |
| Total | 61 | 100.0 | 100.0 | |

The HCOs responded that pharmaceutical sales representatives usually approach doctors directly without going through the hospital management. The hospital management is therefore in favor of a more centralized distribution of promotional funds with a code that forbids direct dealings.

*3.2. Pharmaceutical Companies and Associations*

The four pharmaceutical companies follow guidelines that have been issued by their parent companies. The overall answer is that these house-internal codes are in line with MEACPP already; i.e., they are supportive of the idea of introducing the code. Their fear is that they act code-compliant, but their local competitors have a competitive advantage by being completely free in their promotions. However, this fear might be a bit overblown when considering that local companies are usually far smaller and companies such as Novartis top the lists in most counties that require the publication of payments.

All pharmaceutical companies said that they do not give personal gifts to HCPs and HCOs.

Financial support is mainly given for:

1. Local and international conferences
2. Free samples
3. Product reminders
4. Social gatherings

None of the pharmaceutical companies saw the potential of analyzing the published data and identifying KOLs. One might argue that they do not see the potential of competitive analysis yet, or that they foresee that only around 20% of HCPs will participate. Therefore, the statistics do not cover the whole market.

## 4. Discussion

There seems to be no real resistance against the MEACPP. According to our research, international pharmaceutical companies have such codes in place and already follow them, even when it is not required, such as in MEA. HCPs see no real advantage (e.g., building trust) or disadvantage of such a transparency code. However, one could argue that the MEACPP is drafted in a way that makes it easy to comply with. No publication of financial support given to HCPs and HCOs is mandatory. The example of Germany has shown that when it comes to the crunch of publishing their own data, only 20% of the HCPs allow their data to be publicly visible. Our research leads to the conclusion that the MEA figures will not be much different and might also stabilize at around 20% in the long-term. In other words, around 80% of HCPs and HCOs will not disclose their financial benefits.

It is questionable whether a transparency system makes sense where only 20% of the people are willing to be transparent and 80% are not. Whereas the European pharmaceutical industries and associations argue that the system is working, and it might well be that the opposite is true. The US system, which imposes a law instead of self-regulation has also proven to be ineffective because it has not reduced promotional spending nor increased trust.

In most MENA countries, the governments are not powerful in the sense that they can push through laws easily. Lobbying by the pharmaceutical industry might also play its part in rather compromising on a watered-down self-regulation instead of a strict law on promotional practices in the pharmaceutical industry. A working self-regulating regime is more cost effective than a governmentally imposed law. However, what seems to work in the food industry is not working in the pharmaceutical one. Further research should investigate the reasons why self-regulation works in some industries but not in others.

The main question is why both the US system and the European one have had no effect. Publishing data is good but if patients are not aware of it and politicians do not analyze it and act on the findings, then it is more of a transparency ritual than a real political instrument. Payments can too easily be disguised as research contributions or conference invitations. While disputed, data protection laws are on the side of the HCPs.

A radical solution would be to abandon the idea of the MEACPP altogether. Choosing between introducing or not introducing the MEACPP is like being between a rock and a hard place. The most likely outcome is that it will be introduced within five to 10 years and that 20% of HCPs will participate. Multinational pharmaceutical companies will participate but not many local ones. Associations will stress how effective the new code is, whereas, politicians will take no actions based on the data. Only a small percentage of patients will be aware that such data exists but rarely retrieve how much their own HCP received —even though it will not contribute to more or less trust between the HCP and the patient. The percentage that pharmaceutical companies spend on HCPs and HCOs will increase year by year more than the GDP growth. Areas with the highest payments will be the ones with the highest margin, such as antibiotics, oncologics, and antidiabetics. Pharmaceutical companies will continue nominating HCPs as experts and pay a remuneration for 'research', although they are not really interested in their expertise. They want to make them feel important. Research contracts and invited talks are just bonding tools. Companies will be slow in understanding the potential of the publicly available data to identify KOLs and targeting them accordingly.

There are limitations to our research. Firstly, the number of people who are familiar with MEACPP is small and so is our sample. Even a senior official from a health ministry was unaware of the code. It is difficult to forecast the real disclosing behavior of HCPs. Moreover, since there is no existing system in place, we had to ask hypothetical questions. Cultural aspects were not analyzed. It may well be that Middle Eastern and African countries do not value transparency as much as Western ones.

## 5. Conclusions

The originality of our research is the analysis of challenges related to the transparency in the MEA pharmaceutical industry. We predict that MEACPP will be implemented as self-regulation and will have no effect. This is because MEA governments (health ministries) feel the necessity to follow the Western example of regulating promotions in the pharmaceutical industry. However, experience in the U.S.A. and Europe has shown that neither a binding law nor self-regulation has any effect and patients are rarely aware of it. Pharmaceutical companies will continue spending billions of USD and EUR on promotions. The practical implication is that these two top-down systems do not work. Alternatively, one could think of a bottom-up system whereby HCPs voluntarily refuse to take any payments. This would require a whole new movement and a catchy new slogan, e.g., 'Clean HCP'.

**Author Contributions:** Both authors contributed equally.

**Conflicts of Interest:** The authors declare no conflicts of interest.

## Appendix A

Interview Questions for HCOs incl. trade associations

Are you aware of The Middle East and Africa Code of Promotional Practices in the pharmaceutical industry?
Yes/No
(If not, give short background information on the Code incl. US law and European self-regulation)
What do you personally think of the Code (MEACPP)?

In the USA and Europe, pharma companies must report in detail how much money they give to HCPs and HCOs. This information is available to anybody. I think this is a good idea and should be implemented in MEA. Likert scale 1–5.

Some people say that the pharma industry is corrupt. What is your opinion?

Considering that the data collected are publicly available to anyone, I (we) would be interested in seeing the data from competitors i.e., to see how much money they received from pharma companies.

How much would you/your organisation be willing to spend on an analysis of your competitors (how much they give to which hospitals)/year? _____________ US$

The world becomes more globalised and regulations harmonised, when do you think that ME and Africa will follow the European example and require pharma companies to report financial contributions to HCP/HCO? In how many years from now?
In _______ years or never ( ) ?

Are there other important aspects in the context of MEACPP and US/European pharma regulations which were not covered in the interview?

Interview Questions for Pharma companies

Are you aware of The Middle East and Africa Code of Promotional Practices in the pharmaceutical industry?

(If not, give short background information on the Code incl. US law and European self-regulation)

What do you personally think of the MEACPP?

How does your company currently work with HCP or HCO?
Main projects, e.g., clinical studies, consultancies, speakers, etc.
1. Clinical phase I-III studies
2. Non-interventional studies

3. Phase IV studies (interventional)
4. Advisory boards
5. Internal trainings
6. Other consultancy agreements
7. Speaker arrangements
8. Travel invitations to conferences
9. Others

Does your company have or plan to have a task force or project to implement a system to conform to the Code?

In the USA and Europe, pharma companies must report in detail how much money they give to HCPs and HCOs. This information is available to anybody. I think this is a good idea and should be implemented in MEA. Likert scale.

Some people say that the pharma industry is corrupt. What is your opinion?

Considering that the data collected are publicly available to anyone, I would be interested in seeing the data from my competitors i.e., to see how much money they paid to HCPs.

We, as a pharma company, would consider this type of information useful for analysis of key opinion leaders (KOL).

How much would you/your organisation be willing to spend on an analysis of your competitors/year? _______________ US$

The world becomes more globalised and regulations harmonised, when do you think that ME and Africa will follow the European example and require pharma companies to report financial contributions to HCP/HCO? In how many years from now? In _______ years or never.

Are there other important aspects in the context of MEACPP and US/European pharma regulations which were not covered in the interview?

Questionnaire for HCPs

Are you aware of The Middle East and Africa Code of Promotional Practices in the pharmaceutical industry? Yes () No ()

Some people say that the pharma industry is corrupt. What is your opinion?
Totally disagree (1), disagree (2), neutral (3), agree (4), totally agree (5)

In the USA and Europe, pharma companies must report in detail how much money they give to HCPs and HCOs. This information is available to anybody. I think this is a good idea and should be implemented in MEA.
Totally disagree (1), disagree (2), neutral (3), agree (4), totally agree (5)

Considering that the data collected are publicly available to anyone, I would be interested in seeing the data from my colleagues i.e., to see how much money they received from pharma companies.
Totally disagree (1), disagree (2), neutral (3), agree (4), totally agree (5)

How much would you be willing to spend on an analysis of your colleague's income from pharma companies/year? _______________ US$

If pharma companies were asked to report the financial contributions (gifts, conference invitations etc.) that I have received, then I will insist on my privacy (data protection) so they cannot publish my data.
Totally disagree (1), disagree (2), neutral (3), agree (4), totally agree (5)

The world becomes more globalised and regulations harmonised, when do you think that ME and Africa will follow the European example and require pharma companies to report financial contributions to HCPs/HCOs? In how many years from now? In _______ years or never ().

For me, transparency and trust go hand in hand i.e., if the financial contributions that an HCP/HCO receives get published then the patients will have more trust in the HCP/HCO.
Totally disagree (1), disagree (2), neutral (3), agree (4), totally agree (5)

Patients will not be interested in searching what amount of money his/her HCP/HCO received.
Totally disagree (1), disagree (2), neutral (3), agree (4), totally agree (5)

Plus Demographics

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
