# Peer review of "The Middle East and Africa Code of Promotional Practices in the Pharmaceutical Industry"

_admsci, doi:10.3390/admsci8030053_

Round 1

Reviewer 1 Report

The authors present a research about the Code of Promotional Practices in the Pharmaceutical Industry in the Middle East and Africa. It is an interesting field of research for this category of studies and for the pharmacy industry in general.

However, the study should be improved on the following issues:

The abstract must be structured again. The purpose of the research, the methodology carried out and the design of the research must be shown. Also the findings and limitations.

This phrase must be deleted:"Being the first scholarly paper on the topic". 

The authors can refer to the research gap but not say that it is the first article published in this category because it is not specifically so.

The last paragraph of the introduction should contain the objectives of the research and present the research structure.

Table 1 must be changed. Its quality can also be improved. In its current form, it is not possible to read it. This is mandatory to reach a high-quality study.

Figure 1 should be improved. The quality is not enough and the content cannot be read.

In the conclusion part, the authors should indicate the Practical implications of the study. What practical/professional and academic consequences will it have for the future? In addition, authors should make references to what is the originality and value of the research work.

This paragraph, or similar, should appear in the abstract: 

"Interviews were conducted with pharmaceutical companies and associations. HCPs filled in a questionnaire. "

Section 3 ”Materials and Methods "has insufficient literature review. It must be improved, this is mandatory to reach a high-quality study.

The questionnaire and interview questions should appear in a table as well as the studies from which they have taken the questions.

Author Response

We thank the reviewers for their valuable comments!

REVIEWER 1:

The abstract must be structured again. The purpose of the research, the methodology carried out and the design of the research must be shown. Also the findings and limitations.

We restructured and added: The purpose of this research is the evaluation of the Middle East and Africa Code of Promotional Practices as preliminary draft and its implications.

The limitation of our research is the fact that the MEACPP has not been implemented yet and survey results are therefore based on expectations rather than real events.

This phrase must be deleted: "Being the first scholarly paper on the topic". 

Deleted.

The authors can refer to the research gap but not say that it is the first article published in this category because it is not specifically so.

It actually is. But we happily deleted it.

The last paragraph of the introduction should contain the objectives of the research and present the research structure.

We added: The objectives of the study are:

- Assess the willingness of HCPs to allow pharmaceutical companies to publish personalized data on the financial support given.

- Estimate how many years it will take to get the MEACPP implemented.

- Evaluate the pros and cons of a law versus self-regulation.

- Predict which model MEA will adopt and likely outcomes. 

We moved the research framework up. It is now part of the introduction.

Table 1 must be changed. Its quality can also be improved. In its current form, it is not possible to read it. This is mandatory to reach a high-quality study.

We deleted Table 1 because the official report forms are all of the same poor quality and omitting parts would distort the whole picture of the report.

Figure 1 should be improved. The quality is not enough and the content cannot be read.

We cut and enlarged the focal point and added some interpretation of the data.

In the conclusion part, the authors should indicate the Practical implications of the study. What practical/professional and academic consequences will it have for the future? In addition, authors should make references to what is the originality and value of the research work.

We added under Conclusions:

The originality of our research is the analysis of challenges related to the transparency in the MEA pharmaceutical industry. We predict that MEACPP will be implemented as self-regulation and will have no effect. This is because MEA governments (health ministries) feel the necessity to follow the Western example of regulating promotions in the pharmaceutical industry. However, experience in the U.S.A. and Europe has shown that neither a binding law nor self-regulation have any effect and patients are rarely aware of it. Pharmaceutical companies will continue spending billions of USD and EUR on promotions. The practical implication is that these two top-down systems do not work. Alternatively, one could think of a bottom-up system whereby HCPs voluntarily refuse to take any payments. This would require a whole new movement with a catchy slogan e.g. ‘Clean HCP’.

This paragraph, or similar, should appear in the abstract: "Interviews were conducted with pharmaceutical companies and associations. HCPs filled in a questionnaire. "

Done. Added in abstract.

Section 3 ”Materials and Methods "has insufficient literature review. It must be improved, this is mandatory to reach a high-quality study.

We added:

Fabbri, A, Santos, A, Mezinska, S, Mulinari, S, & Mintzes, B 2018, 'Sunshine Policies and Murky Shadows in Europe: Disclosure of Pharmaceutical Industry Payments to Health Professionals in Nine European Countries', International Journal Of Health Policy & Management, Vol. 7, No. 6, pp. 504-509.

Global Pharmaceuticals Industry Profile 2017, Pharmaceuticals Industry Profile: Global, pp. 1-37, Business Source Premier, EBSCOhost, viewed 28 August 2018.

Jiho, Y, Rosales, C, & Talluri, S 2018, 'Inter-firm partnerships - strategic alliances in the pharmaceutical industry', International Journal Of Production Research, Vol. 56, No. 1/2, pp. 862-881.

Lakdawalla, DN 2018, 'Economics of the Pharmaceutical Industry', Journal Of Economic Literature, Vol. 56, No. 2, pp. 397-449.

We would like to point out that there is no literature on MEACPP. On the web there are a few powerpoint presentations with superficial content. 

The questionnaire and interview questions should appear in a table as well as the studies from which they have taken the questions.

We added: We interviewed health ministries as well as people who were part of the initial drafting of the MEACPP. In addition, we corresponded with authors who have published on the Sunshine Act (U.S.A.) and the German transparency code. Based on these interviews we developed our quantitative questionnaire. Since there was no existing questionnaire we developed our own based on these interviews.

Since interview guidelines and questionnaires are relatively long we placed them into the appendix.

Reviewer 2 Report

First of all let me state that I read the paper reviewed with a great pleasure, not only as a researcher, but also as a consumer of health services.  I’d like to congratulate the Author (s) as the topic raised in the said paper is really very important nowadays. And probably it will be much more important in the future.

When reviewing scientific papers for publication, I usually start with a general overview in terms of a structure, abstract, literature review, methodology, findings of the research, discussion,  conclusions, as well as limitations of the study. I also pay attention to a language level, especially if the paper is written in English, and English is not the native language.

The reviewed paper entitled: “The Middle East and Africa Code of Promotional Practices in the Pharmaceutical Industry” has been analyzed in original submission. The paper is organized in the following paragraphs: 1. Introduction, 2. Literature Review, 3. Materials and Methods, 4. Findings and Discussion, 5. Conclusions, References.

I permit to suggest the authors that the article should be amended as follows:

 1. articles or papers in Administrative Sciences MDPI Journal usually start with “Introduction” and go along with “Materials and Methods”, “Results”, “Discussion”, “Conclusion (not mandatory)”, ” and so on. There is no firm standard way of writing an article, anyway the article should be readable by people in this field; this reviewer would like to refer to other articles in this journal;

2.      the order of the paragraphs as specified above should be followed;

3.      section 2 should be rearranged into: “Introduction” and section 4 in “Results” and “Discussion”;

4. section “Materials and Methods” should be better described. What are the research hypotheses? What method of sample selection was used? In my opinion, the section can not end with a table but its description;

5. The information in Table 1 is completely illegible;

6. Figure 1 is not legible.

7. Figure 3 may be incomprehensible. In my opinion, it is better to present the results on a pie chart.

8. The literature review is quite good and is strongly founded in the existing literature of the topic (22 papers).  Generally I claim that Author (s) provide solid theoretical foundations for the analysis using appropriate references. I would, however, recommend to add some references devoted to the  latest literature associated with the topic in question (including SCOPUS and Web of Science papers).

In conclusion, the authors should modify the manuscript. Then, it will be easier to understand the contents, the results, highlighting the objective and the relevance of the research paper. What is the power of generalisation of the conclusions and  benefit from this work? 

I am eager to read the new version of the paper, which is both novel and appealing.

Author Response

We thank the reviewers for their valuable comments!

REVIEWER 2:

 1. articles or papers in Administrative Sciences MDPI Journal usually start with “Introduction” and go along with “Materials and Methods”, “Results”, “Discussion”, “Conclusion (not mandatory)”, ” and so on. There is no firm standard way of writing an article, anyway the article should be readable by people in this field; this reviewer would like to refer to other articles in this journal;

We changed the structure in 1. Introduction, 2. Materials and Methods, 3. Results, 4. Discussion, and 5.Conclusion. We cited one Admin. Sci. article. 

2.      the order of the paragraphs as specified above should be followed;

Done.

3.      section 2 should be rearranged into: “Introduction” and section 4 in “Results” and “Discussion”;

Done.

4. section “Materials and Methods” should be better described. What are the research hypotheses? What method of sample selection was used? In my opinion, the section can not end with a table but its description;

We added: Our research hypotheses are that (1) HCPs will oppose the idea of publicizing their received contributions. (2) Pharmaceutical companies will also not be enthused by the idea of making their payments transparent. (3) Pharmaceutical associations will show that they support transparency in order to avoid legal sanctions but being representative of the industry they will try to keep public disclosure of financial data to a minimum. (4) Patients will be interested in retrieving the amount of contributions their practitioner received. (5) Health Ministries will be the driving force in pushing transparency codes forward.

We added following description under the Table: In most countries the top three medical disciplines where doctors work are: General Medicine, Internal Medicine and Pediatrics. However, the authors feel that the discipline does not have an impact on the survey results. Any profession is using medication by pharmaceutical companies. The real question is less about the discipline but the volume that a doctor is prescribing.   

We used convenience sampling whereby HCPs were encouraged to spread the word about the survey amongst their friends.

5. The information in Table 1 is completely illegible;

We deleted Table 1 because the official report forms are all of the same poor quality and omitting parts would distort the whole picture of the report.

6. Figure 1 is not legible.

We cut and enlarged the focal point.

7. Figure 3 may be incomprehensible. In my opinion, it is better to present the results on a pie chart.

Replaced with pie chart.

8. The literature review is quite good and is strongly founded in the existing literature of the topic (22 papers).  Generally I claim that Author (s) provide solid theoretical foundations for the analysis using appropriate references. I would, however, recommend to add some references devoted to the  latest literature associated with the topic in question (including SCOPUS and Web of Science papers).

We added:

Fabbri, A, Santos, A, Mezinska, S, Mulinari, S, & Mintzes, B 2018, 'Sunshine Policies and Murky Shadows in Europe: Disclosure of Pharmaceutical Industry Payments to Health Professionals in Nine European Countries', International Journal Of Health Policy & Management, Vol. 7, No. 6, pp. 504-509.

Global Pharmaceuticals Industry Profile 2017, Pharmaceuticals Industry Profile: Global, pp. 1-37, Business Source Premier, EBSCOhost, viewed 28 August 2018.

Jiho, Y, Rosales, C, & Talluri, S 2018, 'Inter-firm partnerships - strategic alliances in the pharmaceutical industry', International Journal Of Production Research, Vol. 56, No. 1/2, pp. 862-881.

Lakdawalla, DN 2018, 'Economics of the Pharmaceutical Industry', Journal Of Economic Literature, Vol. 56, No. 2, pp. 397-449.

We would like to point out that there is no literature on MEACPP. On the web there are a few powerpoint presentations with superficial content.

In conclusion, the authors should modify the manuscript. Then, it will be easier to understand the contents, the results, highlighting the objective and the relevance of the research paper. What is the power of generalisation of the conclusions and  benefit from this work?

Done.

I am eager to read the new version of the paper, which is both novel and appealing.

Our pleasure.

Round 2

Reviewer 1 Report

The authors have made the indicated changes

Reviewer 2 Report

Dear Authors,

Thank you for the new version of the paper.

The authors correctly executed all of the reviewer's indications in a short period of time.

My decision is: accept.

Best regards